# Semi-Supervised Learning for Molecular Graphs via Ensemble Consensus

## Abstract

Machine learning is transforming molecular sciences by accelerating property prediction, simulation, and the discovery of new molecules and materials. Acquiring labeled data in these domains is often costly and time-consuming, whereas large collections of unlabeled molecular data are readily available. Standard semi-supervised learning methods often rely on label-preserving augmentations, which are challenging to design in the molecular domain, where minor changes can drastically alter properties. In this work, we show that semi-supervised methods that rely on an ensemble consensus can boost predictive accuracy across a diverse range of molecular datasets, task types, and graph neural network architectures. Notably, we show that training with an ensemble consensus objective results in an effect similar to knowledge distillation; an individual member of an ensemble trained this way often outperforms a full ensemble trained in a traditional supervised fashion. In addition, this type of semi-supervised training reduces calibration error and is robust over different datasets.

## 1 Introduction

In recent years, machine learning has emerged as a transformative tool in the molecular sciences, accelerating discovery in areas ranging from predicting quantum mechanical properties (Schütt et al., 2021; 2017; Musaelian et al., 2023; Wood et al., 2025) to discovering novel drugs (Wong et al., 2024; Kellenberger et al., 2007; Vidler et al., 2013; Zhuang et al., 2014; Ren et al., 2023) and catalysts (Pillai et al., 2023; Sun et al., 2024; Bai et al., 2025). However, despite recent efforts to curate large labeled datasets (Merchant et al., 2023; Levine et al., 2025), the scarcity of labeled data remains a fundamental bottleneck.

In materials and drug discovery, labels often come from computationally expensive simulations, such as density functional theory (DFT), or resource-intensive laboratory measurements. Consequently, datasets with specialized high-quality labels are typically small, while large databases of unlabeled molecules (e.g., ZINC (Irwin et al., 2012; Kim et al., 2024)) are not fully exploited. This scenario—abundant unlabeled data coupled with scarce labeled data—is an ideal setting for semi-supervised learning (SSL).

Yet, many state-of-the-art methods are poorly suited for the molecular domain. Dominant techniques such as consistency training (Berthelot et al., 2019; Sohn et al., 2020) critically depend on data augmentation strategies that create perturbed copies of an input while preserving its label. Such augmentations are notoriously difficult to design for molecules, where minor structural changes can drastically alter the chemical properties we aim to predict. Meanwhile, approaches such as iterative pseudo-labeling (Scudder, 1965; Riloff & Wiebe, 2003; Huang et al., 2022) hinges on the ability to reliably rank predictions by confidence in order to select the best candidates for pseudo-labeling and to avoid reinforcing model errors. This highlights a critical gap where standard SSL benchmarks and algorithms do not translate well to the practical challenges of molecular science.

In this work, we build upon a class of SSL methods that does not require the explicit design of data augmentations, but rather relies on an *ensemble consistency loss*. Specifically, we train a model ensemble where each member learns from labeled data using a standard supervised loss and from unlabeled data using a loss that promotes agreement among the ensemble members. While ensemble coupling in self-supervised learning has been explored in previous work (Sajjadi et al., 2016; Tarvainen & Valpola, 2018; Platanios, 2018), our formulation is theoretically grounded in an ensemble

loss ambiguity decomposition, trains in a single run, and exhibits a knowledge distillation-like effect that has not previously been discussed. As such, our work makes four core contributions:

1. We provide a theoretical motivation for our specific ensemble-consensus approach based on the formal decomposition of ensemble error, which justifies the consensus as a high-quality, better-than-average supervisory signal.

2. We demonstrate that our method robustly improves predictive accuracy across a wide range of molecular datasets and architectures for both regression and classification.

3. We show a powerful knowledge distillation-like effect, where a single model from our consensus-trained ensemble in almost all cases outperforms an entire ensemble trained in a traditional supervised fashion.

4. We also show that our approach, in comparison to other SSL methods reduces the calibration error and does not harm the prediction accuracy on the unlabeled training data.

## 2 BACKGROUND

**Semi-Supervised Learning** Semi-supervised learning (SSL) is a machine learning paradigm designed for settings with a small amount of labeled data and a much larger amount of unlabeled data. The idea is to leverage the unlabeled data to learn about the underlying structure of the data distribution $p(x)$, which in turn improves the model's ability to learn the mapping from inputs to outputs, $p(y|x)$. Effective SSL methods are typically built upon one or more of the following assumptions:

- **Smoothness Assumption:** If two points $x_1, x_2$ are close in a high-density region of the underlying data manifold, their corresponding labels $y_1, y_2$ should also be close or identical.
- **Cluster Assumption:** The data tends to form distinct clusters, and points within the same cluster are likely to share the same label. This implies that a good decision boundary should lie in the low-density region between clusters.

**Consistency Loss** Consistency regularization is currently the most dominant family of SSL methods. The core idea is that the model's prediction for an unlabeled data point should remain consistent under small perturbations. This directly enforces the smoothness assumption. A successful perturbation or data augmentation is one that explores the local neighborhood of a data point on the manifold without changing its label. The objective is typically formulated as minimizing a distance measure (e.g., Mean Squared Error or KL-Divergence) between the model's predictions for two different augmentations of the same input:

$$\mathcal{L}_{\text{consistency}} = \mathbb{E}_{x_u \sim X_u}[D(f_\theta(\text{aug}_1(x_u))||f_\theta(\text{aug}_2(x_u)))].$$

Different choices of the perturbations give rise to a wide range of methods. $\Pi$-models (Sajjadi et al., 2016) enforce that two predictions should be the same under transformations to the data, the use of dropout and random pooling for perturbations to the model. Each unlabeled datapoint is passed through the network twice and penalized for the difference in the predictions between the passes. The benefit of consistency loss is highly linked to the quality of the data augmentation techniques, as shown in (Xie et al., 2020). Temporal ensembling (Laine & Aila, 2017) builds upon this by maintaining an exponential moving average of predictions for each unlabeled example to create a more stable consistency target. Instead of applying a temporal averaging over the predictions, the mean-teacher method (Tarvainen & Valpola, 2018) averages the model weights and uses the predictions of that model as the consistency target. In the above works, the predictions can be seen as coming from a sort of pseudo-ensemble. As the members of this pseudo-ensemble are based on the trajectory or perturbation of a single network, the diversity of the predictions is reduced and biased, which reduces the prediction accuracy as we later highlight.

This problem can be mitigated by introducing multiple different initial weightings of the same architecture and training them in parallel to use as consistency targets. Chen et al. (2021) (cross pseudo supervision) proposes to do this for pixel-wise segmentation, where the prediction of each of the two ensemble members is hard labeled and used as the consistency target. Filipiak et al. (2022) further extends this for pixel-wise segmentation by using $n$ ensemble models and taking all

combinations of hard labeled predictions as the consistency targets. Another paper that explores different ensemble predictions is Platanios (2018). Here the ensemble members are restarted multiple times during training, and the consensus target is computed from a trainable majority vote or Restricted Boltzmann Machine. All the above methods can be seen as stemming from a broad class of SSL methods that rely on the prediction of an ensemble to guide the training of the individual models to improve predictive accuracy.

In many applications, there exist few or no data augmentations that preserve the label of a data point. Examples include molecules, where the chemical properties can be changed significantly under small changes to the molecule. This restricts the consistency loss methods to only rely on perturbations to the model and not the data. This makes the class of ensemble-based SSL methods well-suited for the problem.

**Pseudo-labeling**  Pseudo-labeling (Yarowsky, 1995; Scudder, 1965; Riloff & Wiebe, 2003), also known as self-training or entropy minimization, is a process where an initial model is trained on the labeled data points and then used to predict labels for a large unlabeled dataset. The primary risk of this method is confirmation bias: if the model generates an incorrect pseudo-label with high confidence, it will reinforce its own mistake during retraining, leading to error propagation. To mitigate this risk, modern SSL methods often integrate more sophisticated frameworks. For example, one uncertainty-aware approach uses a model's evidential uncertainty to estimate the quality of each pseudo-label. This enables an adaptive weighting scheme where high-uncertainty (low-quality) pseudo-labels are given a smaller weight in the loss function, reducing their biasing effect. While this can be effective, such a strategy requires an initial, full training phase on the labeled data before the episodic pseudo-labeling can begin. It also introduces several additional tunable hyperparameters related to its episodic schedule, which require careful tuning (Huang et al., 2022).

**Knowledge Distillation**  Knowledge distillation (Buciluă et al., 2006; Hinton et al., 2015) was proposed as a way of using a complex "teacher" model to transfer its knowledge to a simpler "student" model. Usually, the teacher model is either a model with more parameters or the same model with multiple predictions averaged over multiple augmentations of the input, but the use of an ensemble as the teacher has also been explored (Hinton et al., 2015; Fukuda et al., 2017; Malinin et al., 2019). The transfer of knowledge can be enforced at different levels, such as feature representations (Heo et al., 2019) or intermediate layers (Zagoruyko & Komodakis, 2017). Approaches that match predictions are most closely related to our work. Aligning student and teacher predictions resembles the use of consistency targets in semi-supervised learning, with the key distinction that distillation is typically applied post-hoc, and thus lacks a bootstrapping effect where the teacher also benefits from the student's progress. Furthermore, knowledge distillation is often focused on preserving the uncertainty calibration of the teacher or achieving computational efficiency by deploying the smaller student model instead of the larger one.

## 3  THEORETICAL MOTIVATION

The theoretical motivation for our method is grounded in the formal relationship between an ensemble's performance and that of its individual members. Ensemble performance is governed by a fundamental trade-off between the accuracy of the individual models and the diversity of their predictions. This relationship can be expressed through a loss decomposition, which shows that for any convex loss function, the ensemble's loss is guaranteed to be less than or equal to the average of the individual losses (Wood et al., 2024). This stems from Jensen's inequality and takes the general form:

$$\text{Ensemble Loss} = \text{Average Individual Loss} - \text{Ambiguity} \qquad (1)$$

The ambiguity (or diversity) term is a non-negative quantity measuring disagreement among the members. This decomposition reveals that optimal ensemble performance requires not only accurate individual models but also beneficial diversity.

**Mean Squared Error**  This principle is most clearly illustrated in regression with Mean Squared Error (MSE), where the decomposition is exact and well-established (Krogh & Vedelsby, 1994). For an ensemble of $M$ models $\{f_{\theta_m}\}_{m=1}^M$ with a mean prediction $\bar{f}(x)$, the decomposition is:

$$\underbrace{(y - \bar{f}(x))^2}_{\text{Ensemble MSE}} = \underbrace{\frac{1}{M} \sum_{m=1}^{M} (y - f_m(x))^2}_{\text{Average Individual MSE}} - \underbrace{\frac{1}{M} \sum_{m=1}^{M} (\bar{f}(x) - f_m(x))^2}_{\text{Ambiguity (Prediction Variance)}}. \tag{2}$$

Here, the ambiguity is simply the variance of the predictions around the ensemble mean, providing a clear, label-independent measure of diversity.

**Cross-Entropy** The same principle extends to classification, though the decomposition for Cross-Entropy (CE) loss is more nuanced. Using the geometric mean to average probabilities across the ensemble yields a clean, label-independent decomposition, as in regression (Wood et al., 2024). An exact decomposition is also available for the arithmetic mean:

$$\underbrace{-\mathbf{y} \cdot \ln \bar{\mathbf{f}}}_{\text{Ensemble CE Loss}} = \underbrace{-\frac{1}{M} \sum_{m=1}^{M} \mathbf{y} \cdot \ln \mathbf{f}_m}_{\text{Avg. Individual CE Loss}} - \underbrace{\sum_{c=1}^{C} y_c \ln \frac{\frac{1}{M} \sum_{m=1}^{M} f_{m,c}}{(\prod_{m=1}^{M} f_{m,c})^{1/M}}}_{\text{Ambiguity (Label-Dependent)}}, \tag{3}$$

although here the ambiguity term is explicitly a function of the true label vector $\mathbf{y}$ (where $y_c$ is the true probability of class $c$), making it label-dependent (Wood et al., 2024). Crucially, this ambiguity term is still guaranteed to be non-negative, ensuring that the ensemble loss is always less than or equal to the average individual loss.

Because the ensemble consensus is provably superior to the average individual model, using it as a consistency target for unlabeled data is both effective and theoretically well-justified. In addition, the ensemble prediction will be a useful signal as long as the models are better than random. This suggests the ensemble prediction does not need to incorporate a warm-startup to provide a useful predictive signal, as other works have observed (Tarvainen & Valpola, 2018) and used (Filipiak et al., 2022; Platanios, 2018).

## 4 METHOD

### 4.1 FORMAL DESCRIPTION

We address a standard semi-supervised learning problem with a small set of labeled data, $\mathcal{D}_L = \{(x_i, y_i)\}_{i=1}^{N_L}$, and a large set of unlabeled data, $\mathcal{D}_U = \{u_j\}_{j=1}^{N_U}$. We assume that both datasets are drawn from the same underlying distribution. Our method utilizes a deep ensemble of $M$ models, $\bar{f} = \{f_{\theta_m}\}_{m=1}^{M}$, initialized with different random weights.

The training objective is defined on each model $f_{\theta_m}$ within the ensemble. At each training step, its parameters $\theta_m$ are updated to minimize a composite loss, $\mathcal{L}_m$, which combines a standard supervised signal $\mathcal{L}_{\text{sup}}$ with an ensemble-driven consistency signal $\mathcal{L}_{\text{consistency}}$:

$$\mathcal{L}_m = \mathcal{L}_{\text{sup}}(f_{\theta_m}, B_L) + \gamma \mathcal{L}_{\text{consistency}}(f_{\theta_m}, \bar{f}, B_U), \tag{4}$$

where $B_L$ and $B_U$ are mini-batches of labeled and unlabeled data, respectively, and $\gamma$ is the coupling weight. During training, all models are updated simultaneously by minimizing the sum of their individual losses i.e. $\mathcal{L} = \sum_{m=1}^{M} \mathcal{L}_m$. The first term, $\mathcal{L}_{\text{sup}}$, is the standard task-specific loss for model $f_{\theta_m}$ on the labeled batch, such as mean squared error (MSE) for regression or cross-entropy (CE) for classification. The second term, $\mathcal{L}_{\text{consistency}}$, provides the semi-supervised signal. It is calculated for the model $f_{\theta_m}$ but depends on the outputs of the entire ensemble. For each unlabeled sample $u \in B_U$, a consensus prediction, $\bar{f}(u)$, is computed by averaging the predictions of all $M$ models:

$$\bar{f}(u) = \frac{1}{M} \sum_{m=1}^{M} f_{\theta_m}(u). \tag{5}$$

The Consensus prediction serves as the augmentation-free consistency target for model $f_{\theta_m}$. We penalize the discrepancy between model prediction and the ensemble consensus as

$$\mathcal{L}_{\text{consistency}}(f_{\theta_m}, \bar{f}, B_U) = \frac{1}{|B_U|} \sum_{u \in B_U} D\left(f_{\theta_m}(u), \bar{f}(u)\right). \tag{6}$$

Here, $D$ is a suitable distance metric, for example, the task-specific supervised loss (e.g., L2 or KL-divergence). In practice, when minimizing the loss we detach the gradient through $\bar{f}(u)$, as the consensus prediction is at least as accurate as the individual members' predictions on average (see Appendix 3), ensuring that the ensemble is not encouraged to match the less accurate individual predictions. Note, detaching the gradient has been observed to result in failure cases such as *learner collusion* (Jeffares et al., 2023), but in our experience it does not appear to affect results negatively.

## 4.2 CONSENSUS–DIVERSITY DYNAMICS

Our proposed SSL training scheme directly manipulates the trade-off between accurate individual models and high diversity among them. The unsupervised loss term, $\mathcal{L}_u(x_u) = \mathcal{L}(f_{\theta_i}(x_u), f_e(x_u))$, creates a pull towards consensus by guiding each model $f_{\theta_i}$ to agree with the more stable ensemble prediction $f_e$. This directly reduces the average individual error by providing a high-quality supervisory signal for unlabeled data.

Simultaneously, this pull is counteracted by forces that preserve diversity. Each model begins from a unique random initialization and follows a distinct optimization path due to the stochastic nature of mini-batch SGD. This dynamic allows the models to converge to different solutions in parameter space while still agreeing in function space.

Therefore, our method does not eliminate diversity but rather regulates it. The hyperparameter $\gamma$ in the total loss $\mathcal{L} = \mathcal{L}_l + \gamma\mathcal{L}_u$ serves as a direct control over this balance, allowing us to leverage the unlabeled data to improve individual model accuracy without forcing a complete collapse in diversity.

Another benefit of this continuous learning between models is that we should be less likely get stuck on early bad predictions, as can be the case with many forms of pseudo-labeling. This is because the ensemble targets are "moving" with the ensemble. This can explain why we do not need to warmup the coupling loss.

## 5 EXPERIMENTAL SETUP

We evaluate our method in two settings: First, on a quantum chemistry benchmark to demonstrate its relevance for 3D-geometry-based molecular property prediction, and then across a diverse suite of graph-level tasks to assess its broader applicability. All ensemble members were trained on identical mini-batches of supervised data to simplify implementation. While this strategy reduces ensemble diversity, potentially limiting the ensemble's predictive power, it allows for a fair direct comparison with single models.

**Semi-supervised Protocol**  To simulate the common scenario of data scarcity, we restrict the supervised portion of our training to a small fraction for each task (10%). The remaining training data (90%) is treated as unlabeled and is used exclusively for our ensemble consistency loss. Our primary baseline is a standard deep ensemble of the same architecture, trained only on this small labeled data subset. This setup allows us to directly measure the performance gain from leveraging unlabeled data.

**Datasets**  We test our method on a wide range of different datasets. We perform a prediction of molecular properties in the QM9 dataset (Wu et al., 2018) for the main 12 targets, using the PaiNN architecture Schütt et al. (2021) and an ensemble of size $M = 4$. To investigate how our method scales, we study and compare performance on a single target (internal energy at 0K) for different ensemble sizes ($M \in \{1, 2, 3, 4\}$). For broader validation of our method, we adopt a comprehensive benchmark suite of graph-level tasks. We use three different graph-based architectures: GCN (Kipf & Welling, 2017), GIN (Xu et al., 2019), and GatedGCN (Bresson & Laurent, 2018), adapting the code from Luo et al. (2025) and following the testing procedure from Rampášek et al. (2023). We refer to this suite of benchmarks as GNN+ benchmarks. The ensemble size is fixed to $M = 4$. To demonstrate the general applicability of our method beyond the molecular domain, we perform experiments on a benchmark of non-molecular graph datasets (see Appendix A.3). Further experiments showing broader applicability beyond graphs are included in Appendix A.1. All datasets were split into 80% training data, 10% validation data and 10% test data. While 10% of the training data is used as

Table 1: PaiNN performance (MAE) on QM9 targets. Results are reported as mean $\pm 1.96$ standard error of the mean over 3 seeds.

| Target | Unit | Individual Member | | Ensemble (M=4) | | Mean-teacher |
|---|---|---|---|---|---|---|
| | | Supervised | Supervised + SSL | Supervised | Supervised + SSL | |
| $\mu$ | D | $.0740_{\pm.0009}$ | $.0619_{\pm.0004}$ | $.0683_{\pm.0005}$ | $.0613_{\pm.0004}$ | $.0721_{\pm.0024}$ |
| $\alpha$ | $a_0^3$ | $.1626_{\pm.0006}$ | $.1327_{\pm.0002}$ | $.1423_{\pm.0005}$ | $.1307_{\pm.0002}$ | $.1570_{\pm.0020}$ |
| $\epsilon_{\text{HOMO}}$ | meV | $80.4961_{\pm.5696}$ | $74.3067_{\pm.2874}$ | $76.3702_{\pm.4998}$ | $73.4216_{\pm.2602}$ | $80.6_{\pm2.0}$ |
| $\epsilon_{\text{LUMO}}$ | meV | $62.0410_{\pm.6873}$ | $57.7005_{\pm.3890}$ | $59.3295_{\pm.7226}$ | $57.2248_{\pm.3748}$ | $62.0_{\pm.8}$ |
| $\Delta\epsilon$ | meV | $125.1425_{\pm.7803}$ | $116.8389_{\pm.7476}$ | $119.2652_{\pm.6489}$ | $115.5238_{\pm.7658}$ | $125.0_{\pm2.6}$ |
| $\langle R^2 \rangle$ | $a_0^2$ | $.8141_{\pm.0194}$ | $.6189_{\pm.0321}$ | $.6405_{\pm.0133}$ | $.5696_{\pm.0320}$ | $.799_{\pm.040}$ |
| ZPVE | meV | $2.2163_{\pm.0031}$ | $2.0117_{\pm.0072}$ | $2.0714_{\pm.0036}$ | $1.9884_{\pm.0077}$ | $2.18_{\pm.04}$ |
| $U_0$ | meV | $24.8996_{\pm.2355}$ | $19.9535_{\pm.1812}$ | $20.8927_{\pm.2374}$ | $19.3611_{\pm.1819}$ | $24.7_{\pm.85}$ |
| $U$ | meV | $25.1262_{\pm.3544}$ | $20.1439_{\pm.1736}$ | $21.0568_{\pm.3108}$ | $19.5466_{\pm.1586}$ | $25.0_{\pm1.00}$ |
| $H$ | meV | $25.1391_{\pm.3319}$ | $20.1217_{\pm.1890}$ | $21.0790_{\pm.3171}$ | $19.5235_{\pm.1905}$ | $24.8_{\pm.96}$ |
| $G$ | meV | $25.3738_{\pm.3001}$ | $20.2668_{\pm.2044}$ | $21.3625_{\pm.2596}$ | $19.6880_{\pm.1955}$ | $25.2_{\pm.75}$ |
| $C_v$ | $\frac{\text{cal}}{\text{mol K}}$ | $.0569_{\pm.0005}$ | $.0450_{\pm.0003}$ | $.0491_{\pm.0005}$ | $.0440_{\pm.0004}$ | $.0557_{\pm.0005}$ |

labeled training data, the labels for the remaining 90% are discarded and this data is used as unlabeled (unsupervised) training data.

**Hyperparameter Tuning** To ensure well-tuned models for datasets, the training hyperparameters (learning rate and weight decay) were optimized for each target and model based on the validation performance of a single model in the supervised setting on the reduced labeled data. These hyperparameters were kept fixed across different SSL methods tested to ensure fair comparison. The parameters associated with each specific SSL method (coupling weight, mean-teacher decay, etc.) were optimized based on validation accuracy for each target on QM9, and selected for the GNN+ datasets based on the best value of ZINC. Details about the tuning procedures and selected hyperparameters can be found in Appendix B.

**Evaluation** We evaluate the predictive performance for a single model, a standard ensemble, an ensemble using SSL via ensemble consensus (ours) and an individual member from the latter. All results are reported as the mean along with 1.96 times the standard error of the mean across different seeds.

# 6 RESULTS

## 6.1 MOLECULAR PROPERTY PREDICTION ON QM9

The performance of our method on the 12 regression targets of the QM9 dataset is presented in Table 1. The results indicate that training with the ensemble consistency loss ("Supervised + SSL") reduces the MAE across all evaluated targets when compared to the supervised-only baseline. This is observed for both the individual PaiNN models and the four-member ensembles. Furthermore, n single individual model from the coupled ensemble consistently outperforms the traditional supervised ensemble on all targets.

The results for molecular property prediction on QM9 for different ensemble sizes are shown in Table 2. Our SSL method outperforms a traditional ensemble for all sizes tested. Additionally, using an individual member from an ensemble trained using our proposed SSL method, we not only outperform a standard single model but also perform at a similar level to an ensemble that has only been trained on the supervised data. The results are consistent across all ensemble sizes. Performance increases with more ensemble members.

## 6.2 GNN+ BENCHMARK

To assess the broader applicability of our method, we evaluate it on several molecule-related benchmarks using three different GNN architectures. The results are summarized in Table 3, and are consistent with the performance on QM9. Looking at a single model, the addition of the SSL task

Table 2: PaiNN performance (MAE) on QM9 internal energy at 0K in eV ($U_0$) for different ensemble sizes averaged across 3 seeds, with mean $\pm 1.96$ standard error of the mean.

| | Individual member | | Ensemble | |
| --- | --- | --- | --- | --- |
| Size (M) | Supervised | Supervised + SSL | Supervised | Supervised + SSL |
| 1 | $24.8996_{\pm.2355}$ | – | – | – |
| 2 | – | $20.7268_{\pm.3312}$ | $21.9658_{\pm.6189}$ | $20.2639_{\pm.3139}$ |
| 3 | – | $20.4214_{\pm.2054}$ | $21.2858_{\pm.3761}$ | $19.8955_{\pm.1882}$ |
| 4 | – | $19.9535_{\pm.1812}$ | $20.8927_{\pm.2374}$ | $19.3611_{\pm.1819}$ |

Table 3: Performance on molecule-related benchmarks using different GNN architectures averaged across 5 seeds.

| Dataset | Training | Metric | GCN Individual | GCN Ensemble | GIN Individual | GIN Ensemble | GatedGCN Individual | GatedGCN Ensemble |
| --- | --- | --- | --- | --- | --- | --- | --- | --- |
| ZINC | Supervised | MAE ↓ | $.3163_{\pm.0121}$ | $.2934_{\pm.0094}$ | $.2765_{\pm.0247}$ | $.2516_{\pm.0136}$ | $.2920_{\pm.0113}$ | $.2646_{\pm.0235}$ |
| | Consensus | | $.2406_{\pm.0150}$ | $.2367_{\pm.0148}$ | $.2519_{\pm.0246}$ | $.2485_{\pm.0232}$ | $.2717_{\pm.0230}$ | $.2658_{\pm.0177}$ |
| | Pairwise | | $.2462_{\pm.0108}$ | $.2390_{\pm.0102}$ | $.2500_{\pm.0083}$ | $.2462_{\pm.0092}$ | $.2653_{\pm.0158}$ | $.2597_{\pm.0171}$ |
| | Mean teacher | | $.2884_{\pm.0128}$ | – | $.2791_{\pm.0117}$ | – | $.2830_{\pm.0159}$ | – |
| Peptides-struct | Supervised | MAE ↓ | $.3047_{\pm.0098}$ | $.2932_{\pm.0084}$ | $.2966_{\pm.0067}$ | $.2918_{\pm.0058}$ | $.2994_{\pm.0105}$ | $.2908_{\pm.0101}$ |
| | Consensus | | $.2868_{\pm.0062}$ | $.2866_{\pm.0061}$ | $.2944_{\pm.0072}$ | $.2938_{\pm.0068}$ | $.2854_{\pm.0061}$ | $.2848_{\pm.0068}$ |
| | Pairwise | | $.2933_{\pm.0031}$ | $.2892_{\pm.0029}$ | $.2916_{\pm.0030}$ | $.2901_{\pm.0029}$ | $.2898_{\pm.0042}$ | $.2870_{\pm.0041}$ |
| | Mean teacher | | $.2985_{\pm.0029}$ | – | $.2948_{\pm.0023}$ | – | $.2953_{\pm.0034}$ | – |
| Peptides-func | Supervised | AP ↑ | $.4931_{\pm.0346}$ | $.5105_{\pm.0342}$ | $.4566_{\pm.0224}$ | $.4765_{\pm.0327}$ | $.4289_{\pm.0051}$ | $.4444_{\pm.0200}$ |
| | Consensus | | $.5070_{\pm.0141}$ | $.5160_{\pm.0141}$ | $.4756_{\pm.0180}$ | $.4815_{\pm.0179}$ | $.4509_{\pm.0144}$ | $.4580_{\pm.0062}$ |
| | Pairwise | | $.5055_{\pm.0151}$ | $.5163_{\pm.0150}$ | $.4739_{\pm.0110}$ | $.4811_{\pm.0117}$ | $.4463_{\pm.0067}$ | $.4548_{\pm.0069}$ |
| | Mean teacher | | $.4893_{\pm.0169}$ | – | $.4611_{\pm.0130}$ | – | $.4352_{\pm.0058}$ | – |
| ogbg-molhiv | Supervised | AUROC ↑ | $.7216_{\pm.0193}$ | $.7357_{\pm.0212}$ | $.7329_{\pm.0166}$ | $.7346_{\pm.0165}$ | $.7312_{\pm.0081}$ | $.7341_{\pm.0107}$ |
| | Consensus | | $.7308_{\pm.0218}$ | $.7357_{\pm.0212}$ | $.7339_{\pm.0149}$ | $.7347_{\pm.0153}$ | $.7361_{\pm.0069}$ | $.7383_{\pm.0073}$ |
| | Pairwise | | $.7247_{\pm.0160}$ | $.7336_{\pm.0146}$ | $.7273_{\pm.0128}$ | $.7294_{\pm.0128}$ | $.7375_{\pm.0052}$ | $.7403_{\pm.0050}$ |
| | Mean teacher | | $.7213_{\pm.0161}$ | – | $.6996_{\pm.0207}$ | – | $.7295_{\pm.0165}$ | – |
| ogbg-molpcba | Supervised | AP ↑ | $.1368_{\pm.0025}$ | $.1578_{\pm.0030}$ | $.1421_{\pm.0026}$ | $.1567_{\pm.0029}$ | $.1615_{\pm.0034}$ | $.1779_{\pm.0043}$ |
| | Consensus | | $.1476_{\pm.0023}$ | $.1585_{\pm.0026}$ | $.1496_{\pm.0033}$ | $.1567_{\pm.0039}$ | $.1701_{\pm.0036}$ | $.1781_{\pm.0034}$ |
| | Pairwise | | $.1471_{\pm.0027}$ | $.1597_{\pm.0028}$ | $.1498_{\pm.0021}$ | $.1574_{\pm.0024}$ | $.1674_{\pm.0032}$ | $.1765_{\pm.0034}$ |
| | Mean teacher | | $.1435_{\pm.0016}$ | – | $.1479_{\pm.0037}$ | – | $.1669_{\pm.0028}$ | – |

consistently improves performance over the supervised-only baseline across all datasets and architectures. This performance gain also translates to the full ensembles, which show improvement when trained with the consistency loss. The performance of a single model trained with our SSL method often exceeds that of an entire ensemble trained only on labeled data.

# 7 DISCUSSION

Our experiments on QM9 and the more varied GNN+ benchmark show that our ensemble-based SSL framework consistently improves model performance in low-data regimes. The most significant finding is the substantial boost in accuracy for individual models, a direct result of the knowledge transferred from the ensemble's consensus on unlabeled data. This finding is similar to the idea ensemble distilling (Hinton et al., 2015), where the knowledge of an ensemble is transferred to a single, smaller model, except that our method inherently produces knowledgeable single models. This is explained through the semi-supervised effect on the entire ensemble, resulting in even better ensemble consensus targets for individual models to learn from. This has a key practical benefit: while the method requires an ensemble during training, a single, improved model can be deployed for inference. This offers a valuable trade-off, where an increased one-time training cost yields a final model that is both highly accurate and computationally efficient at inference time. Chemical property screening is a compelling use case, as vast databases of molecules need to be screened resulting in a high inference cost, while the available labeled data and models are small making training cheap. It is noteworthy that for datasets where the parameter related to SSL ($\gamma$ or the mean-teacher decay) was

not directly tuned, the improvement in predictive accuracy was noticeably smaller. This indicates the SSL parameter is highly dependent on the specific dataset.

As shown in Table 2, the predictive performance scales with the number of members in the coupled ensemble. Individual models from the ensemble trained with our method consistently perform at a similar level to an entire traditional ensemble across all ensemble sizes. This finding is further supported in Appendix A.1.

**Limitations** The primary limitation of our approach is the computational overhead associated with training an ensemble consensus model. Transfer learning is another method that is often used in sparsely-labeled settings, which we have not compared against.

**Future work** Our findings suggest several promising avenues for future research. While this work created a semi-supervised split from a fully labeled dataset, a compelling next step would be to use all available labeled data for supervision while introducing a separate, truly unlabeled dataset. This would more directly quantify the benefit of leveraging vast, external chemical libraries and be of interest in a practical setting. Using ensembles for semi-supervised learning also opens the direction for improving accuracy in a principled manner by diversifying the ensemble members through existing techniques. Furthermore, different strategies for how to couple our ensemble can be investigated. While first experiments (Appendix D.4) suggest that a constant coupling weight and an inclusion of the ensemble consistency loss throughout the whole training yield the best results, these findings still need to be validated through further experimentation on a broader range of datasets to find the optimal strategy. Only including the unsupervised data in the later part during training could potentially result in similar predictive performance, while reducing computational cost.

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

SUPPLEMENTARY MATERIAL

## A  EXTENDED STUDIES

### A.1  SCALING WITH NUMBER OF ENSEMBLE MEMBERS

We also investigate the predictive accuracy scaling with the number of ensemble members to larger than 4 sizes. Ensembles of these sizes were not feasible to do on any of the graph datasets, so we instead use the original computer vision version of CIFAR-10. This also validates that our method works for other domains than graphs. We use ResNet-18 (He et al., 2015) with 5,000 labeled and 40,000 unlabeled data-points without any data augmentations. We performed an exhaustive hyperparameter sweep using a single seed over learning rate (0.1, 0.075, 0.05, 0.025, 0.01, 0.0075, 0.005, 0.0025, 0.001), and weight decay (0.01, 0.025, 0.05, 0.075, 0.1, 0.25, 0.5) for the purely supervised model. The number of epochs and learning rate annealing was fixed at a number informally found to work. The parameters of best performing model on validation accuracy at the last epoch was selected. The optimal values can be found in 14. The coupling weight was fixed kept at $\gamma = 1$.

The hyper-parameters can be found in Appendix B.3. From the accuracy results in Table 4 and calibration scores in section A.2, we see a significant increase in accuracy and calibration scores going from a single model to a coupled ensemble with just two models. Interestingly, the individual prediction accuracy of a model trained in a coupled ensemble of two models outperforms the ensemble prediction from all decoupled ensemble sizes tested. This highlights the semi-supervised effect from using unlabeled data for training. Looking at the calibration metrics in Appendix A.2, we see that the calibration results for the coupled ensemble are worse than the uncoupled one. This is often seen in self-supervised learning, as the "self-validating" training can result in worse calibration from confirmation bias (Arazo et al., 2020; Mishra et al., 2024). Surprisingly, we see the individual calibration improving over the decoupled model (i.e., a single model), and also improving as the number of ensemble members increases.

Table 4: Predictive accuracy (%) on CIFAR-10 validation, comparing Decoupled and Coupled models. The values represent mean $\pm$ 1.96 standard error of the mean.

| | Individual Accuracy % | | Ensemble Accuracy (%) | |
| --- | --- | --- | --- | --- |
| Ensemble size | Decoupled | Coupled | Decoupled | Coupled |
| 1 | $59.08_{\pm 1.35}$ | $\cdots$ | $\cdots$ | $\cdots$ |
| 2 | $\vdots$ | $66.36_{\pm 0.45}$ | $62.51_{\pm 0.40}$ | $66.96_{\pm 0.47}$ |
| 4 | $\vdots$ | $67.24_{\pm 0.40}$ | $64.65_{\pm 0.46}$ | $67.92_{\pm 0.49}$ |
| 8 | $\vdots$ | $67.64_{\pm 0.35}$ | $65.73_{\pm 0.51}$ | $68.34_{\pm 0.34}$ |
| 16 | $\vdots$ | $67.75_{\pm 0.32}$ | $66.41_{\pm 0.57}$ | $68.54_{\pm 0.45}$ |
| 32 | $\vdots$ | $67.75_{\pm 0.30}$ | $66.64_{\pm 0.37}$ | $68.52_{\pm 0.35}$ |

## A.2 CALIBRATION METRICS ON CIFAR-10

Table 5: NLL on CIFAR-10, comparing decoupled and coupled models. The values represent mean $\pm$ 1.96 standard error of the mean.

| | NLL | | | |
| | Individual member | | Ensemble | |
| Ensemble size | Decoupled | Coupled | Decoupled | Coupled |
|---|---|---|---|---|
| 1 | $1.543_{\pm 0.109}$ | $\cdots$ | $\cdots$ | $\cdots$ |
| 2 | $\vdots$ | $1.217_{\pm 0.021}$ | $1.267_{\pm 0.020}$ | $1.161_{\pm 0.021}$ |
| 4 | $\vdots$ | $1.169_{\pm 0.019}$ | $1.121_{\pm 0.012}$ | $1.096_{\pm 0.019}$ |
| 8 | $\vdots$ | $1.142_{\pm 0.017}$ | $1.048_{\pm 0.015}$ | $1.064_{\pm 0.016}$ |
| 16 | $\vdots$ | $1.126_{\pm 0.015}$ | $1.007_{\pm 0.015}$ | $1.047_{\pm 0.014}$ |
| 32 | $\vdots$ | $1.123_{\pm 0.019}$ | $0.990_{\pm 0.011}$ | $1.042_{\pm 0.018}$ |

Table 6: AUC-ROC on CIFAR-10, comparing decoupled and coupled models. The values represent mean $\pm$ 1.96 standard error of the mean.

| | AUC-ROC | | | |
| | Individual member | | Ensemble | |
| Ensemble size | Decoupled | Coupled | Decoupled | Coupled |
|---|---|---|---|---|
| 1 | $.8885_{\pm .0075}$ | $\cdots$ | $\cdots$ | $\cdots$ |
| 2 | $\vdots$ | $.9250_{\pm .0020}$ | $.9125_{\pm .0021}$ | $.9292_{\pm .0020}$ |
| 4 | $\vdots$ | $.9295_{\pm .0019}$ | $.9266_{\pm .0016}$ | $.9349_{\pm .0019}$ |
| 8 | $\vdots$ | $.9316_{\pm .0019}$ | $.9336_{\pm .0021}$ | $.9377_{\pm .0018}$ |
| 16 | $\vdots$ | $.9323_{\pm .0016}$ | $.9384_{\pm .0019}$ | $.9386_{\pm .0015}$ |
| 32 | $\vdots$ | $.9329_{\pm .0019}$ | $.9409_{\pm .0017}$ | $.9394_{\pm .0019}$ |

Table 7: ECE on CIFAR-10, comparing decoupled and coupled models. The values represent mean $\pm$ 1.96 standard error of the mean.

| | ECE | | | |
| | Individual member | | Ensemble | |
| Ensemble size | Decoupled | Coupled | Decoupled | Coupled |
|---|---|---|---|---|
| 1 | $.2210_{\pm .0357}$ | $\cdots$ | $\cdots$ | $\cdots$ |
| 2 | $\vdots$ | $.1713_{\pm .0041}$ | $.1128_{\pm .0057}$ | $.1512_{\pm .0049}$ |
| 4 | $\vdots$ | $.1609_{\pm .0034}$ | $.0591_{\pm .0043}$ | $.1369_{\pm .0040}$ |
| 8 | $\vdots$ | $.1548_{\pm .0031}$ | $.0320_{\pm .0043}$ | $.1301_{\pm .0035}$ |
| 16 | $\vdots$ | $.1494_{\pm .0028}$ | $.0243_{\pm .0033}$ | $.1235_{\pm .0030}$ |
| 32 | $\vdots$ | $.1485_{\pm .0044}$ | $.0207_{\pm .0043}$ | $.1226_{\pm .0043}$ |

Table 8: Brier score on CIFAR-10, comparing decoupled and coupled models. The values represent mean $\pm$ 1.96 standard error of the mean.

| | Brier | | | |
|---|---|---|---|---|
| | Individual member | | Ensemble | |
| Ensemble size | Decoupled | Coupled | Decoupled | Coupled |
| 1 | $.4854_{\pm.0271}$ | $\cdots$ | $\cdots$ | $\cdots$ |
| 2 | $\vdots$ | $.5594_{\pm.0042}$ | $.4585_{\pm.0041}$ | $.5530_{\pm.0040}$ |
| 4 | $\vdots$ | $.5654_{\pm.0040}$ | $.4422_{\pm.0047}$ | $.5572_{\pm.0040}$ |
| 8 | $\vdots$ | $.5676_{\pm.0048}$ | $.4316_{\pm.0050}$ | $.5588_{\pm.0047}$ |
| 16 | $\vdots$ | $.5652_{\pm.0044}$ | $.4283_{\pm.0049}$ | $.5563_{\pm.0043}$ |
| 32 | $\vdots$ | $.5649_{\pm.0030}$ | $.4263_{\pm.0033}$ | $.5558_{\pm.0030}$ |

## A.3 NON-CHEMICAL GNN+ DATASETS

Results for non-chemical GNN+ datasets are shown in Table 9. Note the consensus and mean-teacher run for the GatedGCN models were not computed, as the models were too large to fit in memory.

Table 9: Performance on non-molecule-related benchmarks, comparing supervised models with those using additional self-supervised learning (SSL). Results are shown for individual models (Individual) and the full ensemble (Ensemble). Results are the mean $\pm1.96$ standard error of the mean over 5 different seeds.

| | | | GCN | | GIN | | GatedGCN | |
|---|---|---|---|---|---|---|---|---|
| Dataset | Training | Metric | Individual | Ensemble | Individual | Ensemble | Individual | Ensemble |
| CIFAR-10 | Supervised | Acc (%)↑ | $50.44_{\pm0.33}$ | $55.38_{\pm0.49}$ | $50.46_{\pm0.34}$ | $53.90_{\pm0.50}$ | $57.69_{\pm0.34}$ | $61.23_{\pm0.45}$ |
| | Consensus | | $55.33_{\pm0.31}$ | $57.11_{\pm0.42}$ | $54.30_{\pm0.36}$ | $55.60_{\pm0.31}$ | | |
| | Mean teacher | | $50.64_{\pm0.28}$ | | $50.99_{\pm0.86}$ | - | - | - |
| MNIST | Supervised | Acc (%)↑ | $96.61_{\pm0.07}$ | $96.97_{\pm0.04}$ | $96.26_{\pm0.10}$ | $96.73_{\pm0.13}$ | $96.96_{\pm0.05}$ | $97.38_{\pm0.11}$ |
| | Consensus | | $96.82_{\pm0.08}$ | $96.93_{\pm0.11}$ | $96.68_{\pm0.09}$ | $96.82_{\pm0.11}$ | $97.48_{\pm0.06}$ | $97.57_{\pm0.07}$ |
| | Mean teacher | | $96.55_{\pm0.06}$ | - | $96.31_{\pm0.11}$ | - | $96.84_{\pm0.13}$ | - |

## B HYPERPARAMETERS

### B.1 QM9

Our hyperparameter search for QM9 followed a two-step process. First, we started with baseline hyperparameters from a fully supervised setting and tuned the learning rate and weight decay for a single model on the 10% labeled data subset. Second, using these optimized parameters, we then tuned the coupling weight ($\gamma$) for the size-4 ensemble by searching over $\{1.0, 0.1, 0.01, 0.001, 0.0001\}$. The coupling weight swept for the mean-teacher was $\{0.9, 0.95, 0.99, 0.995, 0.999\}$. Final a architectural and training configurations are detailed in Table 10 and Table 11.

Table 10: Hyperparameter Configuration for QM9. These are fixed across all targets.

| Hyperparameter | Value |
| --- | --- |
| **Training** | |
| Batch size | 32 |
| Epochs | 1000 |
| Optimizer | AdamW |
| Scheduler | Cosine annealing |
| **Coupling** | |
| Unsupervised loss criterion | L2 |

Table 11: Additional hyperparameter Configuration for QM9 for different targets.

| Target | Learning rate | Weight decay | Coupling weight | Mean teacher decay |
| --- | --- | --- | --- | --- |
| $\mu$ | 1e-3 | 1e-3 | 0.1 | 0.995 |
| $\alpha$ | 1e-4 | 1e-3 | 0.1 | 0.99 |
| $\epsilon_{\text{HOMO}}$ | 1e-3 | 0 | 0.01 | 0.95 |
| $\epsilon_{\text{LUMO}}$ | 5e-4 | 1e-6 | 0.01 | 0.9 |
| $\Delta\epsilon$ | 1e-3 | 0 | 0.01 | 0.99 |
| $\langle R^2 \rangle$ | 5e-4 | 1e-4 | 0.1 | 0.99 |
| ZPVE | 5e-4 | 1e-5 | 0.001 | 0.99 |
| $U_0$ | 1e-4 | 1e-4 | 0.01 | 0.99 |
| $U$ | 1e-4 | 0 | 0.01 | 0.9 |
| $H$ | 1e-4 | 1e-4 | 0.01 | 0.9 |
| $G$ | 1e-4 | 1e-5 | 0.01 | 0.995 |
| $C_v$ | 1e-4 | 1e-5 | 0.01 | 0.995 |

## B.2    GNN+ DATASETS

We keep the hyperparameters for the different datasets and models the same as in the original paper, except for the number of epochs, weight decay, and learning rate. As we are training with $10\%$ of the original data, we double the number of epochs to mitigate the fewer parameter updates. We then made a two-step hyper-parameter sweep; initially the learning rate using original weight decay values, and afterwards the weight decay using the found best learning rates. The learning rates investigated were $(0.25, 0.5, 1.0, 2.0, 4.0)$ times the original learning rate value for that model and dataset. The weight decays investigated was $(10^{-6}, 10^{-5}, 10^{-4}, 10^{-3}, 10^{-2}, 10^{-1}, 0)$. We could not simply multiply the weight decay values by a fixed factor, as some of the original weight decay values were 0. These sweeps were performed for a single uncoupled model following the same tuning procedure as in the original paper. Notably, this means that the predictive accuracy report from each run is the best validation performance seen during any of the epochs. The found learning rates are listed in Table 13, and weight decays Table 13 below. The train, validation, and test splits follow the same procedure as Luo et al. (2025). Each seed shuffles the labeled and unlabeled part of the training data.

The SSL parameters were selected based on the best performing values on the validation score on ZINC. The mean-teacher values investigated was $(0.9, 0.99, 0.995, 0.999)$, and the coupling weight for the consensus and pair-wise methods were $(0.25, 0.5, 0.75, 1, 1.25, 1.5, 1.75, 2.0)$. The optimal value of mean-teacher was found to be 0.999, and coupling weight for the consensus learning was 1.0, and the pairwise loss was tied between 0.5 and 0.75, so we went with 0.5 based on the recommendations in Filipiak et al. (2022).

Table 12: Tuned learning rates for GNN models across datasets.

| Dataset | GCN | GINE | GATEDGCN |
|---|---|---|---|
| CIFAR-10 | 0.002 | 0.0005 | 0.001 |
| CLUSTER | 0.0005 | 0.0005 | 0.002 |
| ogbg-molhiv | 0.0001 | 0.00005 | 0.0004 |
| MalNet-Tiny | 0.00025 | 0.002 | 0.002 |
| MNIST | 0.001 | 0.002 | 0.001 |
| PATTERN | 0.004 | 0.001 | 0.000125 |
| ogbg-molpcba | 0.000125 | 0.000125 | 0.00025 |
| peptides-func | 0.0005 | 0.002 | 0.002 |
| peptides-struct | 0.002 | 0.0005 | 0.002 |
| ogbg-ppa | 0.0006 | 0.0012 | 0.0003 |
| PascalVOC-SP | 0.004 | 0.002 | 0.0005 |
| ZINC | 0.004 | 0.001 | 0.004 |

Table 13: Tuned weight decays for GNN models across datasets.

| Dataset | GCN | GINE | GATEDGCN |
|---|---|---|---|
| CIFAR-10 | $10^{-2}$ | $10^{-1}$ | $10^{-2}$ |
| CLUSTER | 0 | $10^{-1}$ | $10^{-6}$ |
| ogbg-molhiv | $10^{-3}$ | $10^{-1}$ | $10^{-5}$ |
| MalNet-Tiny | $10^{-4}$ | $10^{-2}$ | $10^{-4}$ |
| MNIST | $10^{-1}$ | $10^{-2}$ | $10^{-5}$ |
| PATTERN | $10^{-3}$ | $10^{-2}$ | $10^{-1}$ |
| ogbg-molpcba | $10^{-1}$ | $10^{-2}$ | $10^{-5}$ |
| peptides-func | 0 | $10^{-1}$ | $10^{-3}$ |
| peptides-struct | $10^{-3}$ | $10^{-5}$ | $10^{-1}$ |
| ogbg-ppa | $10^{-1}$ | $10^{-1}$ | $10^{-2}$ |
| PascalVOC-SP | $10^{-1}$ | $10^{-4}$ | $10^{-2}$ |
| ZINC | $10^{-1}$ | $10^{-5}$ | $10^{-3}$ |

## B.3 CIFAR-10

The hyperparameter configurations for CIFAR-10 are shown in Table 14.

Table 14: Hyperparameter Configuration for CIFAR-10.

| Hyperparameter | Value |
| --- | --- |
| **Learning Rate** | |
| Learning rate | 0.005 |
| Annealing method | Step |
| Step size | 1 |
| Learning rate reduction | 0.975 |
| **Regularization** | |
| L2 Weight Decay | 0.075 |
| **Optimizer** | |
| Optimizer | SGD |
| Momentum | 0.9 |
| **Training** | |
| Epochs | 250 |
| **Loss Function** | |
| Coupled loss weighting | 1.0 |
| Ensemble coupled loss | KL-divergence |
| Supervised loss | Cross-entropy |

## C  CALIBRATION SCORES FOR THE OGBG-MOLHIV

We also investigate the calibration on the ogbg-molhiv benchmark. We do not investigate the datasets ogbg-pcba and peptides functional due to the to the large skewing of classes and missing values. The results are included in Table 15 and Table 16. We see across different architectures that the coupling of the ensemble improves the calibration scores, especially NLL. One notable exception is the MCE score for the GIN ensemble model, where the coupled ensemble becomes significantly worse.

Table 15: Individual Performance on the ogbg-molhiv dataset

| | GCN | | GIN | | GatedGCN | |
| --- | --- | --- | --- | --- | --- | --- |
| Metric | Decoupled | Coupled | Decoupled | Coupled | Decoupled | Coupled |
| Accuracy | $95.78_{\pm 0.38}$ | $96.18_{\pm 0.48}$ | $95.97_{\pm 0.68}$ | $96.30_{\pm 0.35}$ | $95.66_{\pm 0.72}$ | $96.01_{\pm 0.57}$ |
| ROC-AUC | $.721_{\pm .0193}$ | $.731_{\pm .0218}$ | $.733_{\pm .017}$ | $.734_{\pm .015}$ | $.731_{\pm .008}$ | $.736_{\pm .007}$ |
| NLL | $.375_{\pm .185}$ | $.230_{\pm .0662}$ | $.147_{\pm .015}$ | $.140_{\pm .012}$ | $.200_{\pm .033}$ | $.180_{\pm .023}$ |
| ECE | $.0312_{\pm .0092}$ | $.0246_{\pm .0039}$ | $.0113_{\pm .0045}$ | $.0105_{\pm .0048}$ | $.0232_{\pm .0069}$ | $.0201_{\pm .0049}$ |
| MCE | $.2041_{\pm .0994}$ | $.2058_{\pm .0763}$ | $.1113_{\pm .0620}$ | $.1058_{\pm .0246}$ | $.1154_{\pm .0399}$ | $.0985_{\pm .0287}$ |

Table 16: Ensemble Performance on the ogbg-molhiv dataset

| | GCN | | GIN | | GatedGCN | |
| --- | --- | --- | --- | --- | --- | --- |
| Metric | Decoupled | Coupled | Decoupled | Coupled | Decoupled | Coupled |
| Accuracy | $96.66_{\pm 0.33}$ | $96.60_{\pm 0.20}$ | $96.11_{\pm 0.66}$ | $96.39_{\pm 0.33}$ | $96.03_{\pm 0.62}$ | $96.12_{\pm 0.56}$ |
| ROC-AUC | $.7350_{\pm .0228}$ | $.7357_{\pm .0212}$ | $.7346_{\pm .0165}$ | $.7347_{\pm .0153}$ | $.7341_{\pm .0107}$ | $.7383_{\pm .0073}$ |
| NLL | $.2437_{\pm .1051}$ | $.1760_{\pm .0275}$ | $.1432_{\pm .0130}$ | $.1383_{\pm .0108}$ | $.1821_{\pm .0249}$ | $.1729_{\pm .0208}$ |
| ECE | $.0261_{\pm .0057}$ | $.0224_{\pm .0046}$ | $.0121_{\pm .0039}$ | $.0109_{\pm .0037}$ | $.0201_{\pm .0051}$ | $.0193_{\pm .0045}$ |
| MCE | $.2587_{\pm .0793}$ | $.2617_{\pm .0564}$ | $.1585_{\pm .0760}$ | $.1933_{\pm .0852}$ | $.1566_{\pm .0576}$ | $.1533_{\pm .0251}$ |

# D ABLATION STUDIES

## D.1 SOFT OR HARD LABELS FOR CLASSIFICATION

Often semi-supervised methods use some form of "hard-labeling" as the consistency target. Usually, this is implemented as setting the ensemble target for an unlabeled datapoint to be the most likely label, as predicted by the individual model (Filipiak et al., 2022; Tarvainen & Valpola, 2018) or the ensemble (Platanios, 2018). This removes the underlying uncertainty information of the estimates, and risking drastically reducing the calibration of the model by making it overconfident. The motivation for using hard-labeling is the assumption of label smoothness, as it forces the model to pick the same label for data points close together. We investigate this assumption in table 17. The results on accuracy show that hard-labelling slightly benefits the accuracy, it comes at the cost of worse calibration metrics such as ECE and MCE for the individual models. The reason for such a small increase in accuracy can be explained by the label-smoothens assumption can be violated for graphs and especially molecules.

Table 17: Calibration metrics on graph CIFAR-10.

| Metric | Non-Ensemble | | Ensemble | |
| --- | --- | --- | --- | --- |
| | Mean | Hard Label | Mean | Hard Label |
| Accuracy (%)↑ | $56.0220_{\pm.2233}$ | $56.2020_{\pm 0.5595}$ | $56.7640_{\pm 0.2742}$ | $57.1920_{\pm 0.4124}$ |
| ROC ↑ | $.9040_{\pm.0017}$ | $.8936_{\pm.0025}$ | $.7598_{\pm.0015}$ | $.7621_{\pm.0022}$ |
| F1 ↑ | $.5586_{\pm.0021}$ | $.5607_{\pm.0051}$ | $.5661_{\pm.0023}$ | $.5706_{\pm.0034}$ |
| ECE ↓ | $.1514_{\pm.0030}$ | $.3034_{\pm.0052}$ | $.4324_{\pm.0027}$ | $.4281_{\pm.0041}$ |
| MCE ↓ | $.2307_{\pm.0030}$ | $.4252_{\pm.0141}$ | $.4324_{\pm.0027}$ | $.4281_{\pm.0041}$ |

## D.2 PAIRWISE OR COUPLED ENSEMBLE

There is a strong theoretical connection between the pairwise loss between ensemble members used in n-CPS and the coupled ensemble loss presented in this work. For a convex loss $\mathcal{L}$ that can be written on the form $\mathcal{L}(x - y)$, then Jensen's inequality yields

$$
\begin{aligned}
L\big(f_{\theta_i}(x) - \mathbb{E}_m[f_{\theta_m}(x)]\big) &= L\big(\mathbb{E}_m[f_{\theta_i}(x) - f_{\theta_m}(x)]\big) \\
&\leq \mathbb{E}_m[L\big(f_{\theta_i}(x) - f_{\theta_m}(x)\big) \\
&= \frac{1}{M}\sum_{m=1}^{M} L\big(f_{\theta_i}(x) - f_{\theta_m}(x)\big) \\
&\leq \frac{1}{M-1}\sum_{m=1}^{M} L\big(f_{\theta_i}(x) - f_{\theta_m}(x)\big).
\end{aligned}
$$

As $f_{\theta_i}(x) - f_{\theta_m}(x) = 0$ if $i = m$ this upper bound is exactly the n-CPS loss. In general this upper bound is not tight, but if $M = 2$ and $\mathcal{L}$ is of the form $(x - y)^l$, e.g. the $l_1$ or $l_2$-loss we get

$$
\begin{aligned}
\mathcal{L}(f_{\theta_1} - \mathbb{E}_m[f_{\theta_m}(x)]) &= \Big(f_{\theta_1} - \frac{f_{\theta_1} + f_{\theta_2}}{2}\Big)^l \\
&= \frac{1}{2^l}(f_{\theta_1} - f_{\theta_2})^l.
\end{aligned}
$$

We see that the two losses are equal up to a scaling factor that disappears if we tune the learning rate.

## D.3 ROBUSTNESS OF COUPLED WEIGHTING

To investigate the robustness of the coupled weighting $\gamma$, we followed the same experimental setup on CIFAR-10 with a Resnet18 model. The results can be seen in Figure 1. From the figure, we see that the validation accuracy is somewhat flat as soon as $\gamma > 1$, but there is a small optimum around $\gamma = 6$. This illustrates that at least for CIFAR-10, the choice of $\gamma$ is robust.

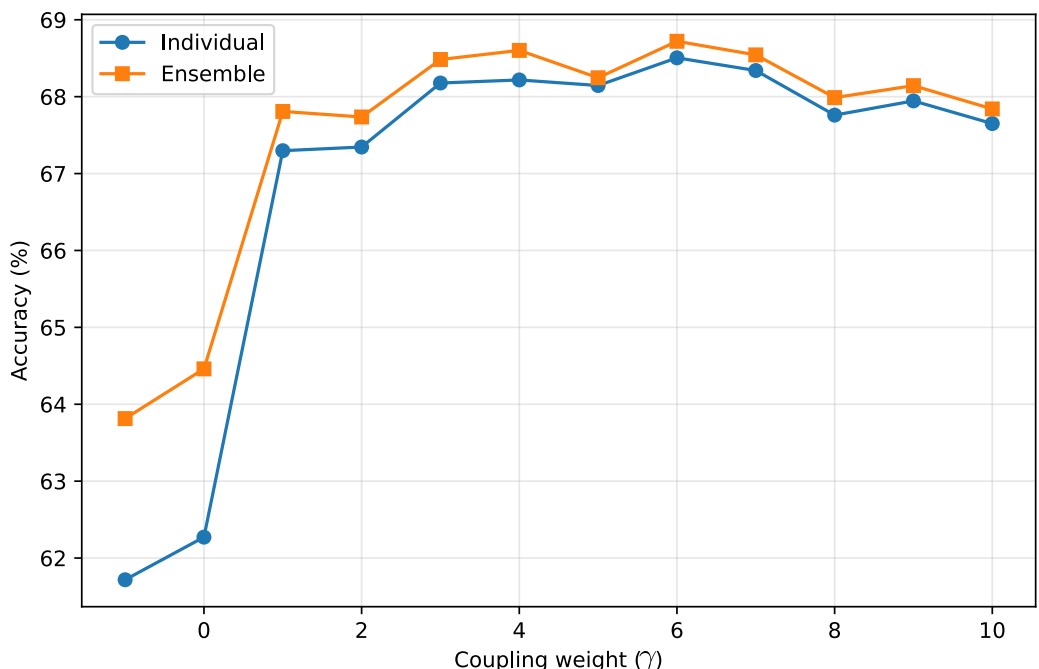

Figure 1: Validation accuracy as a function of the weighting of the ensemble consistency loss.

### D.4 How to Schedule the coupled loss

Initially, during training, the members of the ensemble models only have weak prediction strength. This results in the ensemble prediction serving only as a weak signal guiding the models. Intuitively, this suggests that the weighting of the coupled loss should be added or increased as training progresses. We investigate if this is the case in the same CIFAR-10 setting. We let the ensemble coupling weighting be a linear function of the number of epochs, and vary the starting value and slope of the ensemble coupling weighting. The results can be seen in Figure 2, where negative coupling weights are clipped to 0, while Figure 3 shows the un-clipped results (in the relevant area). From Figure 2, we see that for CIFAR-10, there is no large benefit to begin coupling later compared to selecting a good constant coupling value. Note that a delayed start corresponds to a negative start value and a positive increase pr. epoch, as an initial coupling of -1 and a pr. epoch increase of 0.1 means it starts at epoch 10, due to clipping.

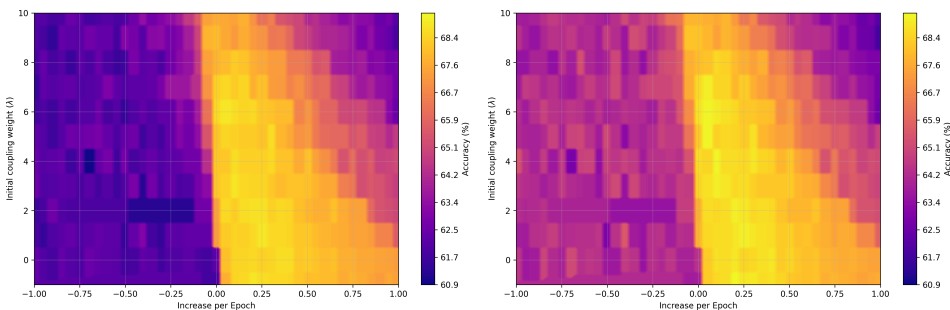

Figure 2: Validation accuracy as a function of the initial coupling weight and the increase in coupling weights per epoch for an individual model (left) and a coupled ensemble with two members (right). The results are averaged over 3 seeds.

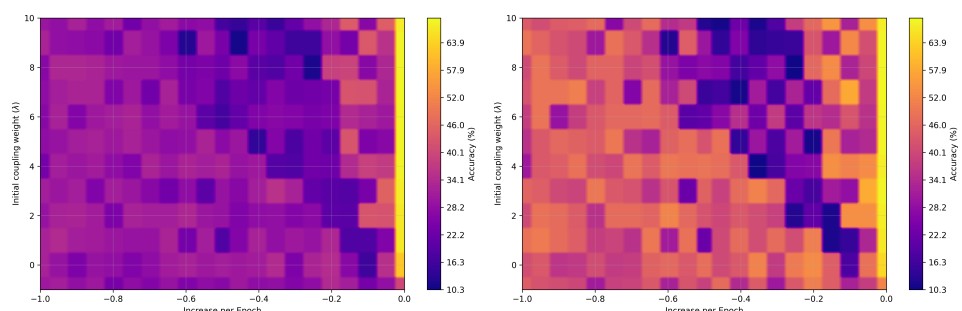

Figure 3: Validation accuracy as a function of the weighting of the ensemble consistency loss.

## D.5 DIFFERENT LOSSES

We also investigated the sensitivity to different formulations of the ensemble consistency loss. The results are shown in Table 18. We ran with the same setup for the computer vision CIFAR-10 and two ensemble members. While the best performing loss function was KL-divergence (the same form as the supervised loss), the "regression" functions ($L_1, L_2, L_\infty$) performed about the same. Only the reversed KL-divergence, $D_{KL}(E||I)$, resulted in lower accuracy, at around the same level as a decoupled model (see Table 4).

Table 18: Validation accuracy with different ensemble consistency loss functions. Results averaged over 10 seeds. Here, $I$ is the individual prediction and $E$ is the ensemble consensus.

| Ensemble Loss | Individual Accuracy |
|---|---|
| $L_\infty$ | $66.23_{\pm 0.29}$ |
| $D_{KL}(I||E)$ | $66.62_{\pm 0.51}$ |
| $D_{KL}(E||I)$ | $59.37_{\pm 0.78}$ |
| $L_1$ | $66.01_{\pm 0.51}$ |
| $L_2$ | $66.12_{\pm 0.45}$ |

## D.6 DIFFERENT COUPLING STRATEGIES

We investigated different strategies for coupling the unsupervised loss on QM9. This includes various combinations of three parameters: the *coupling weight*, the *coupling start* and the *coupling schedule*.

**Coupling weight** The coupling weight parameter defines how much the unsupervised loss should contribute to the total loss. When set to 0, only the supervised loss will be taken into account.

**Coupling start** The coupling start refers to when the unsupervised loss in included during training, i.e. for the first $x\%$ of epochs, the model is only trained on the labeled data and only afterwards, the unsupervised loss with be included via coupling. Depending on the dataset and task, it intuitively can make sense to first let the model learn a little bit before evaluating the loss on unlabeled data. Specifically, in regression tasks this can be the case, since the model output is not bounded, as opposed to classification tasks. When set to 0, coupling will be used through the whole training. This parameter is given in percentage, i.e. percentage of total training epochs after which the coupling should start.

**Coupling schedule** Three different coupling schedules were tested: *constant*, *increase* and *bell*. *Constant* refers to the the coupling weight being constant from onset until the end of training. *Increase* means that the there will be a smooth ramp up until the coupling weight reaches its maximum (i.e. the coupling weight parameter). *Bell* means that there is a smooth bell curve over the coupling weight, i.e. first in increases, then decreases. Here, it will start and end at 0, and peak at a maximum which is set via the coupling weight parameter.

Figure 4 and Figure 5 shows the impact of different coupling strategies on the model performance, here for target 4 and 7 of QM9 respectively. We can see that a good choice of the coupling weight is crucial for our method to result in a significant improvement in MAE compared to the fully supervised baseline. The optimal coupling weight seems to differ per task, as both targets have a different optimum (0.1 for target 4 and 0.01 for target 7). A good value for the coupling start seems to depend on the choice of coupling weight, however a trend can be observed that for the best coupling weight options for each target, the optimal coupling start is 0, i.e. using coupling from the start of training. The optimal choice of coupling schedule seems to depend on both of the other choices, but in the specific case of target 4, the *increase* schedule led to the best performance. For target 7, the *bell* schedule resulted in the best ensemble performance, while the *constant* schedule led to the best individual performance.

One interesting finding here is that if we couple too strongly, meaning we are weighing the unsupervised loss to high, the ensemble performance gets worse than the baseline, while at the same time the individual members from the ensemble are outperforming the baseline. This is due to the models collapsing, so while each individual model is better than an individual model that was not coupled, ensembling has no significant benefit anymore.

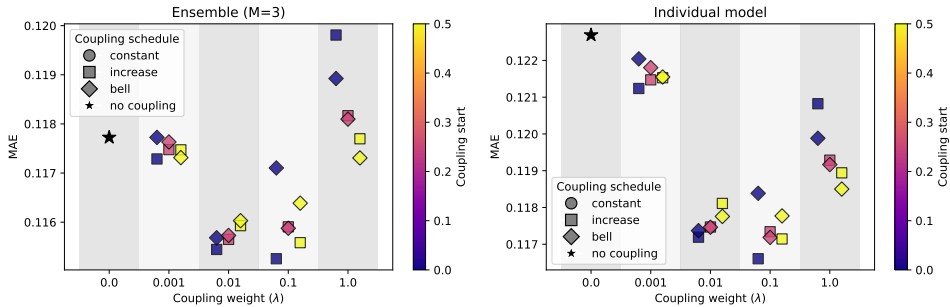

Figure 4: Performance (MAE) of coupled ensembles (left) and individual models from coupled ensembles (right) for different coupling strategies, for QM9 target 4.

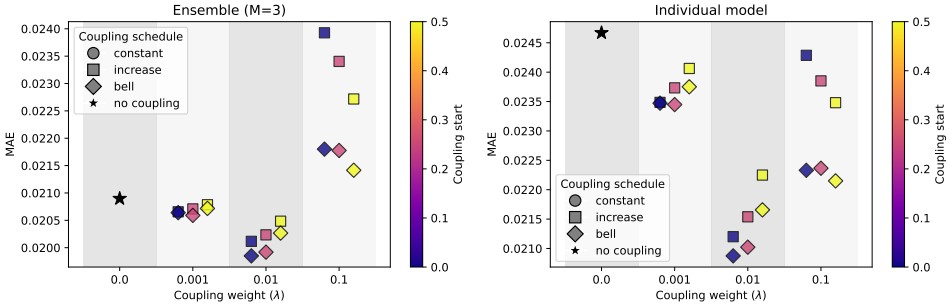

Figure 5: Performance (MAE) of coupled ensembles (left) and individual models from coupled ensembles (right) for different coupling strategies, for QM9 target 7.

### D.7    EVALUATING OVERFITTING ON UNLABELED DATA

To evaluate potential overfitting to the unlabeled data, we compare the final model's performance on the unlabeled training set against its performance on the unseen test set. For this analysis, we leverage our access to the ground-truth labels of the unlabeled set to compute its MAE. As presented in Table 19, the performance is nearly identical across both datasets for all 12 QM9 targets. This strong correspondence indicates that our method avoids overfitting to the unlabeled data used during training. This has a significant practical benefit, as it means the model's predictions on the entire unlabeled set can be reliably used for downstream tasks.

Table 19: PaiNN performance (MAE) on QM9 targets, comparing the held-out test set with the unlabeled dataset used during training. Results are reported for 3 seeds.

| Target | Unit | Data | Individual Member | Ensemble (M=4) |
|---|---|---|---|---|
| $\mu$ | D | Test | $.0619_{\pm.0004}$ | $.0613_{\pm.0004}$ |
| | | Unlabeled | $.0597_{\pm.0005}$ | $.0596_{\pm.0005}$ |
| $\alpha$ | $a_0^3$ | Test | $.1327_{\pm.0002}$ | $.1307_{\pm.0002}$ |
| | | Unlabeled | $.1271_{\pm.0003}$ | $.1264_{\pm.0003}$ |
| $\epsilon_{\text{HOMO}}$ | meV | Test | $74.3067_{\pm.2874}$ | $73.4216_{\pm.2602}$ |
| | | Unlabeled | $71.9862_{\pm.3985}$ | $71.9587_{\pm.3975}$ |
| $\epsilon_{\text{LUMO}}$ | meV | Test | $57.7005_{\pm.3890}$ | $57.2248_{\pm.3748}$ |
| | | Unlabeled | $56.8535_{\pm.2745}$ | $56.8405_{\pm.2741}$ |
| $\Delta\epsilon$ | meV | Test | $116.8389_{\pm.7476}$ | $115.5238_{\pm.7658}$ |
| | | Unlabeled | $114.0289_{\pm.4605}$ | $113.9998_{\pm.4630}$ |
| $\langle R^2 \rangle$ | $a_0^2$ | Test | $.6189_{\pm.0321}$ | $.5696_{\pm.0320}$ |
| | | Unlabeled | $.6011_{\pm.0306}$ | $.5643_{\pm.0303}$ |
| ZPVE | meV | Test | $2.0117_{\pm.0072}$ | $1.9884_{\pm.0077}$ |
| | | Unlabeled | $1.9892_{\pm.0065}$ | $1.9849_{\pm.0065}$ |
| $U_0$ | meV | Test | $19.9535_{\pm.1812}$ | $19.3611_{\pm.1819}$ |
| | | Unlabeled | $19.2928_{\pm.2250}$ | $18.9534_{\pm.2191}$ |
| $U$ | meV | Test | $20.1439_{\pm.1736}$ | $19.5466_{\pm.1586}$ |
| | | Unlabeled | $19.5014_{\pm.1851}$ | $19.1608_{\pm.1790}$ |
| $H$ | meV | Test | $20.1217_{\pm.1890}$ | $19.5235_{\pm.1905}$ |
| | | Unlabeled | $19.4933_{\pm.2367}$ | $19.1509_{\pm.2335}$ |
| $G$ | meV | Test | $20.2668_{\pm.2044}$ | $19.6880_{\pm.1955}$ |
| | | Unlabeled | $19.7077_{\pm.2041}$ | $19.3852_{\pm.2030}$ |
| $C_v$ | $\frac{\text{cal}}{\text{mol K}}$ | Test | $.0450_{\pm.0003}$ | $.0440_{\pm.0004}$ |
| | | Unlabeled | $.0443_{\pm.0002}$ | $.0439_{\pm.0002}$ |

