# OpenReview forum: "Semi-Supervised Learning for Molecular Graphs via Ensemble Consensus"
_ICLR.cc/2026/Conference — ICLR 2026 Conference Withdrawn Submission_

### Official Review · Reviewer_RLvp · 2025-10-29

**Soundness:** 2
**Presentation:** 3
**Contribution:** 2
**Rating:** 2
**Confidence:** 3

**Summary:**

The paper proposes an ensemble-consensus SSL for molecular graphs. For instance, train $M$ independently initialized models with (i) standard supervised loss on scarce labels and (ii) an augmentation-free consistency loss that pulls each member toward the ensemble mean on unlabeled data (gradients detached through the consensus).
The approach is motivated by an ambiguity/ensemble loss decomposition (exact for MSE; non-negative ambiguity for CE), arguing the consensus is a better-than-average target.

Empirically (QM9 and GNN+ suite), the method usually improves accuracy vs supervised single/ensemble baselines and yields a distillation-like effect where a single member from the coupled ensemble often matches or beats a fully supervised ensemble.

**Strengths:**

- Practical & simple: addresses the augmentation bottleneck in molecules; one extra loss with a single coupling weight $\gamma$.
- Results across PaiNN (QM9) and multiple GNNs on ZINC/Peptides/OGB; plus non-graph (CIFAR-10) study.
- Ablations strengthen nicely the paper:
   - $\gamma$ robustness in Fig. 1.
   - scheduling Figs. 2/3.: not big gains from elaborate schedules
   - Loss choices (Table. 18 KL, $L_1$, $L_2$, $L_\infty$)
   - Soft vs hard labels
   - Overfitting check in Table. 19.

**Weaknesses:**

- Novelty positioning vs Mean-Teacher / temporal ensembling / cross pseudo-supervision remains under-specified. Please explicitly delineate what is new beyond using average-prediction consensus with detached gradients and joint training.
- Diversity measurements are still absent. Figures suggest diversity collapse at large $\gamma$ (ensemble degrades while members improve), but there are no quantitative diversity metrics (pairwise logit correlation/JSD, disagreement rates) or plots over epochs.
- Detach-gradient ablation is not reported. Given known ''collusion`` issues, it would be valuable to show detach vs no-detach on at least one dataset.
- Calibration nuance: On CIFAR-10, coupled ensembles have worse calibration than decoupled (A.2), while individuals improve. This should be surfaced in the main paper with discussion/temperature-scaling results.
- Baselines for molecules could be stronger: self-supervised pretrain + fine-tune baselines; simple SE(3)-preserving transforms for invariant QM9 targets; uncertainty-aware pseudo-labeling.
- Compute trade-offs: still no wall-clock/FLOPs/memory/energy vs supervised ensemble and vs pretrain+single.
- Hyperparameter transfer: SSL params for GNN+ chosen on ZINC; a small $\gamma$ sensitivity on 1–2 other datasets would solidify robustness.

**Questions:**

1. Please provide a contrast table vs Mean-Teacher, temporal ensembling, cross pseudo-supervision, and post-hoc distillation, and point to which components you introduce (e.g., consensus-as-teacher across independent members, joint single-run coupling, ambiguity-driven justification, detach choice).
2. Can you report member diversity (pairwise correlation/JSD of logits, disagreement rates) vs $\gamma$ and over training? This would concretely explain the ensemble degradation at large $\gamma$.
3. What happens if you backprop through the consensus? Accuracy/diversity/collusion evidence on CIFAR-10 and one molecular dataset would help.
4. Bring representative NLL/ECE/Brier plots/tables into the main paper, and discuss why ensembles calibrate worse on CIFAR-10 under coupling while individuals improve. Include temperature scaling results.
5. Can you add a graph SSL pretraining baseline (e.g., masking/contrastive) and an uncertainty-filtered pseudo-labeling baseline?
6. Please report wall-clock, FLOPs, memory, and energy for i) supervised single, ii) supervised ensemble, iii) coupled ensemble; and highlight inference savings from single-member deployment.
7. What happens if unlabeled data are scaffold-shifted relative to labeled? Does consensus help/hurt?
8. Fix typos: line 310 "n single"

---

### Official Review · Reviewer_uaBw · 2025-10-30

**Soundness:** 2
**Presentation:** 3
**Contribution:** 1
**Rating:** 2
**Confidence:** 4

**Summary:**

This paper focuses on semi-supervised learning on molecular graph, introducing a ensemble-consensus approach where consensus function can guide individual predictions to align with the consensus. The experimental results demonstrate that incorporating the authors' self-supervised algorithm can further enhance the performance of property prediction.

**Strengths:**

- Self-supervised or semi-supervised learning in the molecular domain is a meaningful research topic because there is a large amount of unlabeled data in the scientific field.

**Weaknesses:**

1. The novelty of this paper needs to be confirmed. Semi-supervised learning has been studied for more than a decade. It is a general method and is not restricted to a particular domain. Ensemble consensus is also a traditional topic. To me, this paper feels like it directly applies some basic semi-supervised algorithms to molecules, but there are no noteworthy updates to the semi-supervised idea and ensemble consensus loss.

2. The results of all properties in Table 1 are significantly lower than the standard baselines. The published errors are basically twice those in the original PaiNN paper, not to mention that PaiNN is a relatively old model and newer models, such as Equiformer, have not been considered. In such underfitted experimental results, it is easy to achieve improvements, but they do not have the significance of guiding the training of all models.

3. The relationship between semi-supervised and unsupervised (self-supervised) learning is very close because they both address the problem of how to use unlabeled data to enhance the model's expressive power. However, this paper lacks comparisons with these methods. There are many molecular self-supervised algorithms [1-5].

[1] Pre-training via Denoising for Molecular Property Prediction

[2] Fractional Denoising for 3D Molecular Pre-training

[3] Sliced Denoising: A Physics-Informed Molecular Pre-Training Method

[4] Equivariant Masked Position Prediction for Efficient Molecular Representation

[5] One transformer can understand both 2D & 3D molecular data

**Questions:**

See weaknesses.

---

### Official Review · Reviewer_46fZ · 2025-10-31

**Soundness:** 3
**Presentation:** 2
**Contribution:** 2
**Rating:** 4
**Confidence:** 4

**Summary:**

The paper introduces an ensemble-consensus semi-supervised learning (SSL) method tailored to molecular graph prediction, addressing the challenge of designing label-preserving augmentations in chemistry. Instead of data perturbations or pseudo-labelling, multiple models are trained jointly with a loss that encourages agreement between each model’s prediction and the ensemble mean on unlabeled data. The approach is theoretically motivated by an ensemble loss–ambiguity decomposition, which guarantees that the ensemble consensus has lower expected loss than the average individual model. Empirical results on QM9 and multiple GNN benchmark datasets demonstrate that this coupling substantially improves both individual and ensemble predictive accuracy, often yielding single models that outperform fully supervised ensembles. The method also shows improved calibration and robustness across architectures.

**Strengths:**

* **Originality:** The idea of using ensemble consensus as an augmentation-free semi-supervised signal is conceptually elegant and well-suited to molecular domains where small structural perturbations alter semantics. The paper builds on ensemble theory to ground SSL in a loss-decomposition framework rarely applied in this context.
* **Quality:** Experiments are extensive and rigorous. The method is evaluated across diverse datasets (molecular and non-molecular), multiple architectures (PaiNN, GCN, GIN, GatedGCN), and against strong SSL baselines such as Mean Teacher and Cross Pseudo Supervision. The consistent gains and ablations (varying ensemble size, calibration analysis) suggest robustness.
* **Clarity:** The presentation is generally clear. The theoretical sections correctly motivate ensemble consensus via convex loss decomposition, and the algorithm is described succinctly with minimal hyperparameters.
* **Significance:** The paper contributes an approach that (i) avoids domain-specific augmentations, (ii) provides a form of implicit distillation during training, and (iii) yields single models competitive with full ensembles—important for computational chemistry where inference cost dominates. The technique is general enough to inspire extensions to other domains with scarce labels and abundant unlabeled data.

**Weaknesses:**

While the theoretical motivation is appealing, its current formulation remains **qualitative** rather than quantitative in terms of the classical *bias–variance decomposition*. The “ambiguity” term from ensemble theory corresponds to prediction variance, yet the paper does not make this connection explicit or measure it empirically. This omission limits the theoretical completeness of the work. Specifically:

* There is **no explicit analysis linking ensemble consensus training to bias and variance terms**, even though the approach directly manipulates both (consensus reduces variance; supervised loss controls bias). A formal bias–variance–diversity decomposition, or empirical estimation thereof, would clarify how the method operates as a variance regulariser.
* The paper does not explore the **theoretical or empirical limits** of the approach:
  what happens as the number of ensemble members ( $M \to \infty$ ), or as the coupling weight ( $\gamma \to \infty$ )?  In these limits, one expects collapse to the ensemble mean (loss of diversity), which would bound the achievable improvement.
* The **empirical scaling with $M$ ** is reported up to 32 members (CIFAR-10 appendix) but not analysed: there is no plot or quantitative fit showing the asymptotic saturation of performance or diversity.
* The method’s computational overhead (linear in $M$ ) is acknowledged but not characterised in terms of wall-time or GPU cost, which would be useful for practitioners.

Overall, the paper would benefit from (i) a more explicit connection between ensemble ambiguity and variance reduction, (ii) a discussion or simulation of the large-ensemble limit, and (iii) empirical visualisation of how performance and diversity scale with $M$.

**Questions:**

1. **Bias–variance connection:**
   Could the authors reformulate the ambiguity decomposition explicitly in bias–variance terms, e.g., decomposing the ensemble generalisation error into bias², variance, and noise components? This would clarify the theoretical position of ensemble consensus within classical statistical learning theory.
2. **Empirical exploration of the large-M limit:**
   Please provide an empirical analysis of ensemble size scaling. For example, plot performance (MAE or AUROC), ensemble variance (prediction disagreement), and calibration error versus $M$ for both coupled and decoupled ensembles. Does performance saturate or decline as $M$ increases?
3. **Coupling strength analysis:**
   How does varying $\gamma$ affect ensemble diversity and individual model accuracy? Could the authors report or plot a “diversity vs. accuracy” curve?
4. **Computational cost:**
   What is the practical training overhead (e.g., runtime or GPU hours) relative to a single supervised model? This would contextualise the trade-off between one-time ensemble training and single-model inference.
5. **Limit interpretation:**
   Theoretically, as $M \to \infty$, the ensemble consensus approximates the expected predictor under the training distribution. Do the authors see this as analogous to a Bayesian model average? If so, could this connection be clarified or discussed in the revised version?

A response addressing these points—particularly the empirical *scaling-with-M* plot and a more explicit link to bias–variance decomposition—would significantly strengthen both the theoretical and empirical sections of the paper.

---

### Official Review · Reviewer_3MoT · 2025-11-01

**Soundness:** 3
**Presentation:** 4
**Contribution:** 2
**Rating:** 4
**Confidence:** 4

**Summary:**

This paper introduces a consensus loss framework for semi-supervised learning on molecular datasets. The proposed method trains an ensemble of models using a composite loss that combines standard regression or classification objectives with a consensus term that encourages agreement among ensemble members. This regularization mechanism aims to improve generalization by leveraging unlabeled data without relying on label-preserving augmentations, which are often challenging in molecular domains. The approach is evaluated on the QM9 dataset and a suite of GNN benchmarks, demonstrating improved performance under limited supervision.

**Strengths:**

- **Novelty**: The use of ensemble consensus as a semi-supervised signal is well-motivated and avoids the pitfalls of data augmentation in molecular settings.
- **Practical Relevance**: Addresses a key challenge in molecular machine learning—data sparsity—by effectively utilizing unlabeled data.

**Weaknesses:**

- **Benchmarking Gaps**: The evaluation lacks comparison with classical ensemble methods such as gradient boosting trees (e.g., XGBoost) and random forests. These baselines are well-established and often competitive in low-data regimes. For instance, how does the proposed method compare when trained on the same 10% subset and evaluated on the same test set as in Table 1?
- **Model Selection Ambiguity**: The claim that a single model from the consensus-trained ensemble outperforms the ensemble raises questions about selection criteria and reproducibility. It is unclear how this model is chosen and why it generalizes better.
- **Ensemble Diversity**: Section 4.2 does not sufficiently explain how diversity among ensemble members is maintained during training. This is critical for ensemble effectiveness.
- **Scaling Analysis**: Table 2 explores performance scaling with ensemble size, but does not provide guidance on the optimal number of models or the trade-offs involved.
- **Theoretical Assumptions**: The Limitations section should acknowledge that the theoretical benefits of ensemble learning under consensus loss rely on convexity assumptions, which may not hold in real-world non-convex settings.

**Questions:**

1. What is the selection criterion for the single model that outperforms the ensemble? Is it based on validation performance, random choice, or another heuristic?
2. Why would an individual member outperform the ensemble? Is this effect consistent across datasets and architectures?
3. How is ensemble diversity ensured during training? Are different initializations or data partitions used?
4. What is the optimal ensemble size for balancing performance and computational cost, based on the scaling results in Table 2?
5. Can the authors provide comparative results against classical ensemble methods such as XGBoost or random forest under the same data constraints?

---

### Note · Authors · 2025-12-24

**Comment:**

We would like to sincerely thank the reviews for their time and insightful feedback.
 We did not engage with the review process mainly due to OpenReview’s problems, but also because the reviewers suggested a gap to publication. We would, however, still like to address some of the raised questions and suggestions with a view to a later work.
We summarize several themes identified across the reviews and provide brief responses to specific questions:
1. We recognize the need for a deeper empirical investigation into how the diversity and variance of the ensemble members impact accuracy and uncertainty metrics.
2. We aim to provide a more rigorous theoretical understanding of how ensemble consensus scales as the number of members (M) increases, specifically looking at asymptotic behavior.
3. We acknowledge the importance of positioning this work more clearly in the context of self-supervised learning (SSL). While we maintain that semi-supervised and unsupervised methodologies address distinct paradigms and can be used in tandem, we agree that they are often applied to the same problems. We will incorporate relevant graph SSL pre-training baselines in the next iteration of this work to better situate our contributions within the broader literature.
Responses to Specific Reviewer Concerns:
Reviewer 3MoT
1. The metric used to demonstrate that an individual model can outperform the ensemble is the test predictive accuracy reported in Tables 1 and 3. We have clarified this formulation in our updated discussion.
2. We hypothesize that the observed performance gains stem from a "virtuous cycle": individual models are improved by the ensemble signal, which in turn enhances the quality of the aggregate ensemble prediction.
3. As specified in lines 247–249, ensemble members are trained on identical data mini-batches. As noted in line 296, each model is initialized with different random weights to ensure initial diversity.
4. Known ensemble literature suggests the predictive accuracy will improve with ensemble size, but with diminishing returns. We also observe this in table 2 and 4.  The preferred ensemble size will depend on the computational resources available to the end user, as larger ensemble sizes will be preferred for maximizing the predictive accuracy.
5. Random Forest models have traditionally not been utilized for QM9, as incorporating the 3D structural data integral to the target properties is non-trivial. Furthermore, RFs typically lack the expressivity of Graph Neural Networks (GNNs). We would welcome references to established works investigating RFs on the QM9 dataset. While extending ensemble consensus to non-gradient-based models is a compelling direction, we consider it out of scope for this initial study.

Reviewer 46fZ
1. We agree that work in this direction is interesting and will investigate it the a later work.
2. While some disagreement measures for larger ensembles (M=32) were included in Appendices A.1 and A.2, we will provide clearer visualizations in the next version to improve readability.
3. We agree analysis of the interplay between the strength of supervised and unsupervised signals will be interesting.
4. We will include a detailed plot of computational resource usage as requested.
5. The discussion is interesting. We hypothesize that the ensemble consensus does not approximate the expected predictor, due to non-linearities of the network. We will expand on this in future work. In the case of linear models, we hypothesize Bayesian model averaging to be equivalent in the limit as $ M\rightarrow\infty$.

Reviewer uaBw
1. While we agree the idea is simple, we would have liked to ask the reviewer for examples of similar work. To the best of our ability, we have only found sparse related work as detailed in our related works section.
2. Regarding the performance on QM9. It is expected that accuracy is lower than fully supervised baselines because we utilize only 10% of the available labels. During implementation verification, we achieved a lower validation error than the original PaiNN paper by adopting a cosine annealed learning rate. During the verification of our implementation, we obtain a lower validation error than what is reported in the original PaiNN paper due to the change to cosine annealed learning rate.
We maintain that PaiNN remains a highly competitive model for datasets like QM9, often outperforming newer models. For example, PaiNN obtains a lower mean error on QM9 than the mentioned Equiformer on 6 out of 12 targets (https://arxiv.org/abs/2206.11990)

3. See shared response 3. We thank the reviewer for the thorough list of unsupervised papers.

# Reviewer RLvp
1. We will be more explicit about the differences to related works in future work.
2. We agree this investigation would be of interest, and will include this in further work.
3. We intend to include experiments demonstrating the non-issue of "collusion" risks when back-propagating through the ensemble mean versus the current detached approach.
4. Thank you for noticing this seeming difference. We will put more focus on this discrepancy in a later work.
5. We will add a graph SSL pre-training baseline to provide a more comprehensive comparison.
6. As requested, we will provide a comprehensive plot of computational resource usage compared to standard supervised training.
7. We will look more into this in a later work.

We again thank the reviewers.

**Withdrawal Confirmation:**

I have read and agree with the venue's withdrawal policy on behalf of myself and my co-authors.